# Impact of Chronic Fetal Hypoxia and Inflammation on Cardiac Pacemaker Cell Development

**DOI:** 10.3390/cells9030733

**Published:** 2020-03-17

**Authors:** Martin G. Frasch, Dino A. Giussani

**Affiliations:** 1Department of Obstetrics and Gynecology, University of Washington, Seattle, WA 98195, USA; 2Center on Human Development and Disability, University of Washington, Seattle, WA 98195, USA; 3Department of Physiology, Development & Neuroscience, University of Cambridge, Cambridge CB2 1TN, UK; dag26@cam.ac.uk

**Keywords:** intrinsic heart rate variability, iHRV, fetal programming, cardiac development, sinus node

## Abstract

Chronic fetal hypoxia and infection are examples of adverse conditions during complicated pregnancy, which impact cardiac myogenesis and increase the lifetime risk of heart disease. However, the effects that chronic hypoxic or inflammatory environments exert on cardiac pacemaker cells are poorly understood. Here, we review the current evidence and novel avenues of bench-to-bed research in this field of perinatal cardiogenesis as well as its translational significance for early detection of future risk for cardiovascular disease.

## 1. Introduction

A suboptimal prenatal environment, such as fetal exposure to diminished nutrient or oxygen supply, impacts myocardial development, predisposing affected individuals to cardiometabolic disorders in later life [1,2]. However, currently, there are no direct data during development on the timing of the emergence of the cardiac pacemaker cell synchronization (CPS) nor on the effects of adverse intrauterine conditions on cardiac pacemaker cell development and alterations of CPS, the basis for an altered fetal heartbeat. Interestingly, some of the recent insights into cardiac pacemaker cell development have come from bioengineering. Bioengineering has focused on the creation of biological pacemakers [3,4], the biology of the human pluripotent stem cells differentiating into adult cardiomyocytes [5], as well as the transcriptomics associated with cardiac mature pacemaker cell function at single-cell resolution [6].

The analysis of fetal heart rate variability (HRV) has been the mainstay of noninvasive fetal health monitoring for decades, yet its utility to identify endangered fetuses has remained limited [7,8]. In this review, we will discuss how the CPS provides the subcellular, cellular, and multicellular substrate for the emergence of an intrinsic fetal HRV (iHRV). We hope that expanding research into the iHRV as a proxy to fetal cardiac health will not only help in the diagnosis of fetuses at risk in adverse pregnancy, but that it will also inspire future studies to focus on this aspect of cardiac cellular development and its significance for the early detection of cardiac diseases in later life [9,10].

The heartbeat is generated by the sinoatrial node, a small region in the right atrium of the heart, which comprises approximately 10,000 cells that interact with each other to set the normal cardiac rhythm. The fluctuations in heart rate result from a combination of its own intrinsic variability and the modulation of the pacemaker activity by several neural and endocrine control mechanisms (Figure 1) [11]. Remarkably, the intrinsic fluctuations of the heartbeat are seen not only in the isolated node, devoid of all neural and hormonal inputs, but also at the single pacemaker cell level, isolated from the node [12]. The electrical activity in such a single cell is generated by ions flowing through discrete channels within the cell membrane.

Based on experimental data, in an in silico pacemaker model composed of ~6000 channels, the intrinsic dynamics of the irregular beating seen experimentally in single nodal cells was shown to be due to the (pseudo)random opening and closing of single channels [13].

In addition to synchronization between single-channel activity, pacemaker cells rely on an interstitial tissue architecture for their isolation from the rest of the myocardium and a reservoir of brown adipose tissue for ready access to energy substrates [14,15,16]. This is in contrast to the reliance on glycogen reserves for energy supply in the myocardium.

Together, these three components (Figure 1), composed of ion channels, isolated tissue, and the energy reservoir, form a dynamic system responsible for the generation of the heartbeat and its vulnerability to developmental challenges, such as exposure to hypoxia or infection, which in turn may determine future susceptibility to dysfunction in later life.

The blue box depicts the key intrinsic (cardiac) components of the CPS system as well as the modulatory impact of innervation by the autonomic nervous system’s vagal and sympathetic nerves. During endotoxemia, these extrinsic vagal and sympathetic modulatory influences are altered (green and pink arrows, respectively) by lipopolysaccharide (LPS) molecules on the level of the second messenger systems; in addition, LPS also affects the intrinsic ion channel (I_Ca, T_, I_Ca, L_, I_K_, and I*_f_*) activity, which altogether impacts the CPS. An increase in cAMP concentrations increases HCN4 channel activity, which is blocked by ivabradine as it reduces the I*_f_* (“funny”) ion current.

## 2. How is Cardiac Pacemaker Synchronization (CPS) Achieved?

There is no universally accepted mechanism explaining CPS. However, several important contributing components have been identified.

Each channel in a pacemaker cell has one or more gates, all of which must be open to allow current to flow through the channel. As these gates are thought to open and close in a random manner, each gate can be modeled by a Markov process, assigning a pseudorandom number to each gate every time that it changes state from open to closed or vice versa. This number, in conjunction with the classical voltage-dependent Hodgkin–Huxley-like rate constants that control the speed at which a gate will open or close, then determines when that gate will next change state. In the studies by Guevara and Lewis [13,17], the model’s behavior was consistent with the hypothesis that the irregular beating seen experimentally in single sinus nodal cells is due to the random opening and closing of single channels, and Ca-clock activity (see below for details) was not required in the model.

If one or more electric shocks are delivered directly to the heart near the intrinsic pacemaker, the heart rhythm is modified transiently, but re-establishes itself with the same frequency as before within a few seconds [18]. Such behavior represents a memory of the pattern of the intrinsic CPS, an imprint of the physiological relationship between the pacemaker cells and their dynamic properties, at the channel, cellular, and multicellular levels.

In addition to the complex temporal dynamics of CPS based on the channel physiology and self-organization, there is an emerging role for compartmentalized (i.e., associated with distinct, spatially-confined microdomains) organization of pacemaker signaling complexes in the sinoatrial node pacemaker cells [6]. Upon structural remodeling of the sinoatrial node, disruption in subcellular targeting of pacemaker proteins and associated signaling molecules may affect their biophysical properties, neurohormonal regulation, as well as protein–protein interactions within the pacemaker signaling complex. This will disturb rhythmic generation of action potentials, thereby contributing to the pathophysiology of the sinus node function. Evidence for these relationships is clear, derived from patients and animal models with genetic defects of scaffolding proteins, which are closely associated with sinus node disease; for instance, these defects may occur via indirect changes of key components in the coupled-clock systems in terms of protein expression, function, and membrane localization. This extends beyond the classical concept of electrical remodeling, according to which dysfunction can be explained by straightforward increases or decreases in protein expression alone, and adds a new dimension to cardiovascular disease. It thus introduces a novel framework for therapeutic approaches for the treatment of pacemaker dysfunction, targeted at preventing the degradation of cardiac cytoarchitecture or abnormalities in the compartmentalization of cardiac pacemaker microdomains, incurred as a result of intrauterine insults [19].

A single-cell resolution in situ study of the cardiac conduction system has recently yielded an unprecedented and detailed transcriptional biomarker landscape of the cardiac pacemaker cells that includes ion channels (Hcn4 and Hcn1: hyperpolarization-activated cyclic nucleotide-gated channels 4 and 1), transcription factors (*Isl1, Tbx3, Tbx18, and Shox2*), gap junction gene *Gjc*1, Igfbp5 (insulin-like growth factor-binding protein 5), Smoc2 (SPARC-related modular calcium-binding protein), *Ntm* (Neurotrimin), *Cpne5* (Copine 5), and *Rgs6* (regulator of G-protein signaling type 6) [6,20]. The above represent targets for future studies of CPS development in utero in healthy and complicated pregnancies (Figure 1).

The automaticity of action potentials within the sinoatrial node’s pacemaker cells is driven by a dynamic, bidirectionally coupled interaction of two clock systems. These are represented by the spontaneous rhythmic local Ca^2+^ releases generated by a Ca^2+^ clock and the activity of the electrogenic surface membrane molecules, especially the ion channels (the M clock). The coupling between the intracellular Ca^2+^ and the fluctuations of the surface membrane potential is mediated by the Ca^2+^–cAMP–protein kinase A (PKA) signaling. Consequently, beta-adrenergic receptor stimulation can restore or further enhance this coupling and the CPS, as it increases the cAMP concentrations [10]. The study highlights how dependent CPS is on energy availability and in this sense complements the work by Gu and colleagues [21], discussed further below.

Future developmental studies of the expression profiles of these biomarkers should yield insights into how physiological and pathophysiological stimuli during gestation impact the maturation and synchronization of the cardiac pacemaker cells.

## 3. How is CPS Impacted by Developmental Hypoxia and Infection?

CPS memory goes beyond re-establishing its intrinsic rhythm. Cardiac disease can also impair CPS, leaving its “memory imprint” on it [9]. That is, impaired intrinsic properties of pacemaker cells become manifest in an altered heart rate and HRV in the context of heart disease [9]. Ischemia slows pacemaker activity in rabbit sinoatrial node pacemaker cells because two inward currents, I_Ca,T_ and a presumed I_NCX_, are diminished [22].

Infection, in particular Gram-negative bacterial infection, impacts CPS and therefore HRV via direct effects on pacemaker channels resulting in ionic current remodeling [23,24]. This was studied in the human hyperpolarization-activated cyclic nucleotide-gated channel 2 (hHCN2), which is stably expressed in HEK293 and cardiac pacemaker cells exposed to lipopolysaccharides (LPS). The “funny” current (*I*f) of the pacemaker cells is the main membrane mechanism generating the diastolic depolarization: the phase of the action potential responsible for cardiac spontaneous activity. *I*f is also responsible for the modulation of the heart rate by the autonomic nervous system (ANS) [25]. LPS released from Gram-negative bacteria during severe sepsis reduces both native *I*f and HCN2- or HCN4-mediated currents and shifts their activation to more negative potentials [26]. The LPS effect is not mediated by any of the intracellular modulatory pathways affecting HCN channels, but it is instead mediated by direct interaction with the channel [23]. Klöckner et al. showed that alterations of the HCN current require the integrity of the LPS molecule, as neither the proinflammatory lipid A alone nor the O-chain polysaccharidic region alone was effective. LPS release during sepsis can modify the *I*f properties by shifting them out of their physiological range, which may impair normal rate control and decrease the cardiac safety factor, making the system more prone to life-threatening perturbations (Figure 1) [26].

In a recent study, the vulnerability of sinus node cells to lipotoxicity has been highlighted [27]. Future studies should investigate how this vulnerability relates to fetal exposures to malnutrition, maternal obesity, chronic hypoxia, and intrauterine growth restriction (IUGR).

Energy availability plays a role in CPS via the modulation of the mitochondrial dynamics [21]. Although the data in induced cardiac pacemaker cells should be considered with caution, as it compares to native cardiac pacemaker cells, this approach represents one of the rare opportunities to study this rare subpopulation of human cardiac myocytes with rhythm-generating properties. Isolated, native sinoatrial node cells exhibited a remarkable resilience to prolonged hypoxia followed by reoxygenation [28,29,30], a condition that would irreversibly damage chamber cardiomyocytes. This is due to a significantly lower global metabolic demand in the cardiac pacemaker cells compared to cardiomyocytes. It remains to be investigated whether and from what time point in gestation, from the embryonic stage with tubular heart, on to the chamber formation period and eventually the fully formed fetal heart, the fetal cardiac pacemaker cells exhibit similar ischemia tolerance and what consequences such ischemic events have on programming CPS properties and susceptibility to future insults, e.g., an increased risk of arrhythmia development [31,32].

Overall, ischemia and infection modulate CPS by altering the intrinsic properties of the electrical activity of pacemaker cells. This may lead to changes in HRV and arrhythmia [33]. Three major components with varying degrees of vulnerabilities to adverse intrauterine conditions emerge, namely, HCN2/HCN4 channels, brown adipose tissue, and the intercellular matrix (Figure 1). It is likely that interference by LPS, ischemia, hypoxia, or fibrosis with any of these constituents of the cardiac pacemaker cell environment would yield delays in CPS maturation and function.

## 4. CPS and Fetal Development?

What are the most relevant currents to focus on that contribute to CPS and iHRV [27,34]? The literature suggests that these are the *If*, mediated in particular via HCN4. How do these behave during fetal and early postnatal development?

In the rabbit, the sodium current (I(Na)) contributes to the sinoatrial node automaticity in the newborn, but not in the adult period when heart rate is slower. In contrast, heart rate is high and the I(Na) is functional in the adult mouse sinoatrial node [35]. Canine sinoatrial node cells have TTX-sensitive I(Na), which decreases with postnatal age. This current does not contribute to normal automaticity in isolated adult cells but can be recruited to sustain excitability if nodal cells are hyperpolarized. This is particularly relevant in the fetal and newborn periods, when the I(Na) is large and the autonomic balance favors vagal tone.

The pacemaker current, *I*f, is expressed in ventricular myocytes from neonatal rats and progressively disappears with advancing age. When present, it shows electrophysiological properties similar to *I*f re-expressed in hypertrophied adult rat ventricular myocytes. Thus, it is likely that the occurrence of *I*f in ventricular myocytes of hypertrophied and failing hearts is due to the re-expression of a fetal gene [36].

In the heart, both during development and in adulthood, the kinetic properties and expression levels of *I*f match the need of the cells to perform pacemaker activity. Although at immature stages of embryonic development all cardiomyocytes are autorhythmic and express a robust *I*f current activated within the physiological range of voltages, shortly after birth, the pacemaker activity and *I*f expression become restricted to cells of the conduction system only, whereas in the working myocardium, the current is downregulated and its activation is shifted to more negative, non-physiological voltages [25]. Noteworthy for future studies, HCN4 is expressed in the embryonic chamber myocardium at low levels, making it a suitable marker of sinoatrial node cells already from pre-septation stages.

The characteristics of native *I*f result from a complex interplay between different factors. The kinetic properties and expression levels of the f-channels, encoded by the HCN genes (HCN1–4), are finely regulated by auxiliary proteins (caveolin-3, MiRP-1, KCR-1, SAP97, and TRP8) and lipids (phosphatidylinositol 4,5-bisphosphate) as well as by post-translational modifications, such as phosphorylation and glycosylation. Any alteration of these regulatory pathways may modify the contribution of *I*f to the action potential, leading to an arrhythmogenic phenotype [25].

It is not clear when and how gradually during gestation the CPS develops, so that we can speak of a “healthy” intrinsic pacemaker rhythm. These unknowns, as well as the genetic program of pacemaker cell development, are under investigation [3,37,38]. Our recent data suggest that fetal intrinsic beat-to-beat HRV is present in the healthy fetus near-term and shows a memory of previous exposure to chronic hypoxia [39]. The two-clock coupling mechanism discussed in Section 1 [10] may provide at least a partial explanation for this memory to chronic hypoxic conditions, which, in addition to lowering energy availability, is known to program sympathetic hyperactivity in the offspring [39,40,41].

## 5. What is iHRV?

From studies of healthy adult subjects during exercise and studies of heart transplant patients, the field is aware that intrinsic components of cardiac rhythm can contribute substantially to HRV [11,42,43]. The iHRV is thought to result from a three-scale level system of synchronization occurring at the ion channel, cell, and multi-cell layers of organization [44]. At each of these levels, various physiological and pathophysiological stimuli shape, modulate, and program the synchronization properties and the development of CPS in ways we are only just beginning to unravel. These intrinsic fluctuations at multiple time scales combined with a modulatory input from the ANS explain, at least in part, the fractal and complex properties of iHRV, respectively [11,24,43]. With regard to the modulation by the ANS, the focus so far has been on sympathetic beta-adrenergic signaling. Future work should also explore the modulation by vagal components.

We have recently discovered that iHRV originates in fetal life and that chronic fetal hypoxia significantly alters it [39]. Interestingly, iHRV and its memory of chronic fetal hypoxia are characterized by nonlinear properties reflecting its recurrent state and chaotic behavior, properties inherent to fractal dynamics [45]. The significant relationship between nonlinear measures of fetal iHRV and left ventricular end-diastolic pressure (LVEDP), which is elevated in fetuses from hypoxic pregnancies, suggests that such iHRV indices could reflect fetal myocardial dysfunction, particularly during cardiac diastole. Therefore, alterations in fetal iHRV may prove clinically useful as a biomarker of impaired cardiac reserve and fetal myocardial decompensation during antepartum or intrapartum monitoring of the fetal heart in a high-risk pregnancy or complicated labor.

What are the mechanisms contributing to fetal iHRV in late gestation in a normal pregnancy? The above review of the mechanisms underlying the CPS and influences on its activity suggests a number of factors capable of impacting the ion channel activity directly, such as LPS or hypoxia, via energy availability or via alterations in the compartmentalization architecture of the sinoatrial node pacemaker cells. In Section 1, we describe twelve genes involved in the programmed differentiation of the cardiac pacemaker cells throughout embryonic development. The disruption of this complex spatiotemporal genetic program is likely to impact the CPS and, therefore, leave a pathophysiological imprint on the susceptibility of the cardiac pacemaker cells to future insults, in particular cardiac arrhythmia or arrest. The study of the two-clock coupling-based generation of CPS underscores the intricate interplay between the cardiac pacemaker cells, neural autonomic input, and energy availability as another layer of the putative transfer mechanisms of in utero hypoxia upon iHRV [10]. This likely dependence upon sympathetic influences links fetal adaptation to chronic stressors with the autochthonic responses within the cardiac pacemaker cell network, emerging from the adaptive processes within the excitatory cells themselves in response to chronic hypoxia. Therefore, there are several putative transfer mechanisms by which in utero chronic hypoxia may imprint upon iHRV. A systematic exploration of such candidate pathways would be a rich future avenue of research (summarized in Figure 1).

## 6. Can the Impact of Fetal Hypoxia or Infection on CPS be Captured by HRV Monitoring?

In the above sections, we discuss the contributions to CPS, iHRV, and HRV of the complex, multi-layered activity of the cardiac pacemaker cells. This opens a multitude of questions regarding the clinical application, which we list below in the hope that they will inspire future investigations:(a)Are there suitable HRV properties likely to capture iHRV in vivo? In [39], we report such putative HRV measures representing recurrent states, chaotic, and fractal dynamics. We invite the interested reader to explore this study and test the identified measurable outcomes in their own investigations;(b)Could we distinguish between the effects of chronic fetal hypoxia versus inflammation on CPS from HRV analysis? Our findings ex vivo indicate an impact of chronic hypoxia on CPS and iHRV, and studies in vivo identified HRV signatures capable of tracking fetal systemic and gut- and brain-specific inflammatory responses over a period of days [46,47,48]. However, it is yet not clear to what extent the HRV measures identified in these studies reflect contributions from iHRV. The mechanisms discussed in Section 2 support this possibility. To date, no data exist on the specificity of HRV outcomes to selective challenges, such as cardiac inflammation. This remains the subject of future studies;(c)What is the relationship between CPS and myocardial development? In a recent study, we found a significant correlation between fetal iHRV and two measures of fetal cardiac diastolic dysfunction, namely, LVEDP and the minimum rate of change of ventricular pressure (dP/dt) in fetuses from pregnancies affected by chronic hypoxia [39]. Could this reflect a dual pathological impact of hypoxia on both cardiomyocyte and pacemaker cell development? Alternatively, could these relationships reflect a functional physiological adaptive response relationship between these cell populations? These questions remain open to debate;(d)The HRV code has been proposed as an overarching concept incorporating various multi-scale contributions of interorgan communication reflected in HRV [8,49,50]. We suggest that an understanding of the impact on CPS of chronic fetal hypoxia or infection during pregnancy should be sought within the integrative framework of the HRV code;(e)Could we use mathematical modeling to derive predictions for HRV properties likely to work as iHRV biomarkers in the human clinical setting? Mathematical models exist for CPS and fetal cardiovascular responses to labor, for example [51,52]. Such models could be combined and explored in silico to derive numerical predictions of HRV properties specific to fetal iHRV and their quantitative contribution to fetal cardiac function in vivo.

## 7. Conclusions and Outlook

The three major components of the cardiac pacemaker system—the cellular ion channels, isolated tissue, and energy reservoir—form a dynamic network responsible for the generation of iHRV (Figure 1). Future studies should determine the spatiotemporal vulnerability of this system, i.e., time periods during embryonic and fetal development that are particularly sensitive and susceptible to adverse intrauterine insults, such as chronic hypoxia or infection. This developmental vulnerability, in turn, may determine future susceptibility to cardiac dysfunction or limited tolerance in later life.

Specifically, the following molecular candidate targets could provide interesting studies of CPS development in utero and postnatally: ion channels (HCN4 and HCN1), transcription factors (Isl1, Tbx3, Tbx18, and Shox2), gap junction genes *Gjc1, Igfbp5, Smoc2, Ntm, Cpne5*, and *Rgs6*. Functional studies in pacemaker cells should explore the developmental profile of CPS dynamics in vitro or using the Langendorff preparation ex vivo, for example, in [39], with attention to the vulnerability of the CPS to noxious stimuli such as ischemia, hypoxia, or inflammation. To gauge the impact of energy availability during intrauterine development on CPS maturation, the effects of alterations in nutrition on CPS development should be investigated. Such studies could involve maternal malnutrition or maternal obesity during pregnancy with or without chronic hypoxia and/or infection to assess partial and combined effects. Such studies would also permit the spatial and qualitative characterization of changes in the isolating tissue, such as fibrosis, and the micro-domain organization within the cardiac pacemaker cells.

## Figures and Tables

**Figure 1 cells-09-00733-f001:**
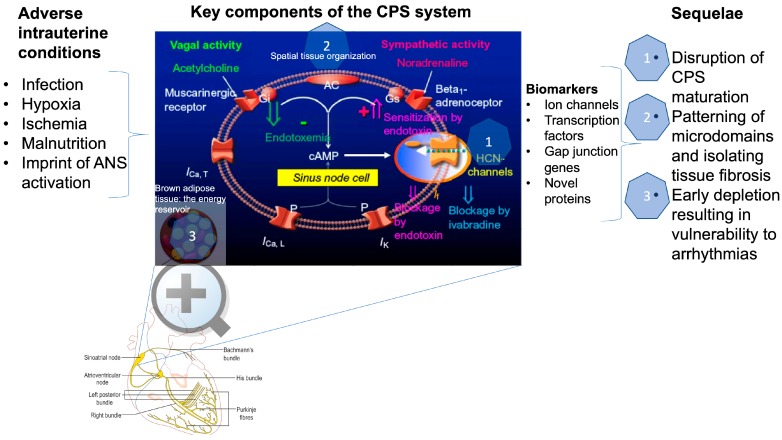
Visual abstract. Adverse intrauterine conditions, such as infection, hypoxia, ischemia, and malnutrition, reprogram the autonomic nervous system (ANS). Together, these adversities impact the three key components of the cardiac pacemaker synchronization (CPS), which are shown in the center blue box (see below for details). The impact is proposed to alter physiology across three conceptual dimensions identified by blue-hued hexagons as follows: (1) direct alteration of the ion channels’ function; (2) change in the spatial organization of cardiac pacemaker cells, i.e., their channel microdomains and the isolating tissue; and (3) reduction in the energy availability in brown adipose tissue. In this review, we discuss known biomarkers that represent the ion channel activity, transcription factors, gap junction genes, and novel proteins, whose role is yet to be understood. These biomarkers can be engaged to quantify the compound effect of a greater vulnerability to future insults. This is identified in the left-hand side under sequelae.

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
