# Peer review of "Impact of Chronic Fetal Hypoxia and Inflammation on Cardiac Pacemaker Cell Development"

_cells, 2020, doi:10.3390/cells9030733_

Round 1

Reviewer 1 Report

This review article broadly covers topics related to the impact of chronical fetal hypoxia and inflammation on cardiac pacemaker cell development. The review is divided in three parts 1) the mechanism leading to the synchronization of cardiac pacemaker cells; 2) the impact of hypoxia and infection on this synchronization; and 3) if the analysis of intrinsic fetal heart rate variability can be used to follow the cardiac pacemaker synchronization. This review is well written, and the authors raised important questions on the vulnerability of this system. However, I have some comments that would help to improve this manuscript. I believe a review in the Cells needs to be accessible to a more general audience of biologists, and as such the authors should make more of an effort to provide relevant background and context for none heart specialists when making their points:

The authors should better explain how the pacemaker is developed, and in particular the three main components of the pacemaker system.

In general, there is no reference to studies describing the development of the pacemaker particularly the regulatory gene network regulating the pacemaker cell identity. Several studies have already reported transcriptome analysis of these pacemaker cells.

It is important for the reader to understand how extrinsic signals can influence the synchronization of the pacemaker cells. Therefore, the authors should better explain the relationships between environment and this system.

The authors did not explain how the significance of heart rate variability (HRV) and how it would affect the development of the fetus.

The author should further develop the part on hypoxia and in particular the pathway that is activated in response to hypoxia.

Figure 1: The illustration is very complex and difficult to understand. The authors should simplify or better explain this Figure.

Author Response

This review article broadly covers topics related to the impact of chronical fetal hypoxia and inflammation on cardiac pacemaker cell development. The review is divided in three parts 1) the mechanism leading to the synchronization of cardiac pacemaker cells; 2) the impact of hypoxia and infection on this synchronization; and 3) if the analysis of intrinsic fetal heart rate variability can be used to follow the cardiac pacemaker synchronization. This review is well written, and the authors raised important questions on the vulnerability of this system. However, I have some comments that would help to improve this manuscript. I believe a review in the Cells needs to be accessible to a more general audience of biologists, and as such the authors should make more of an effort to provide relevant background and context for none heart specialists when making their points:

>> We thank the reviewer for insightful comments and address them below point by point. Edits in the manuscript are highlighted in red. Any new references added in the process of responding to the comments are appended at the bottom of this response.

>> During the peer review process, we received a friendly email comment from a colleague at McGill University whose work we cited in the preprint version of the manuscript, Prof. Michael Guevara. He pointed us to a more recent publication of the relevant work we cited. To provide the readers with the latest developments in the field, we added this publication to the list under Ref. 51 as follows: [1]

The authors should better explain how the pacemaker is developed, and in particular the three main components of the pacemaker system.

>> We review the latest insights into cardiac pacemaker cell development from single-cell transcriptomics on Lines 68-84 and 85-92 noting that the biomarkers of the developmental milestones learned from these studies will be useful for future studies on fetal development of cardiac pacemaker cell synchronization in healthy and complicated pregnancy.

>> Furthermore, we dedicated Section 4 to reviewing the fetal stage of the development of the cardiac pacemaker cell synchronization (Lines 147-183).

>> We characterize the sinoatrial node on Lines 32-49. We illustrate the make-up of the pacemaker and conduction system in Figure 1 (bottom left).

>> Because our focus is on the cardiac pacemaker cell synchronization during fetal development and the impact of adverse events on this process, we abstained from diving into the related topics of the general pacemaker physiology outside of sinoatrial node. We do refer the interested reader to some relevant publications on the topic.

In general, there is no reference to studies describing the development of the pacemaker particularly the regulatory gene network regulating the pacemaker cell identity. Several studies have already reported transcriptome analysis of these pacemaker cells.

>> We appreciate the reviewer’s point. Our focus is on the synchronization of the pacemaker cells and the developmental aspects of this phenomenon. We discuss the pertinent insights from and cite the studies on the transcriptome analysis of the pacemaker cells: Ref 6, 19 and 26 (Lines 24, 85-90, and 127):

>> Ref 6: Goodyer, W.R.; Beyersdorf, B.M.; Paik, D.T.; Tian, L.; Li, G.; Buikema, J.W.; Chirikian, O.; Choi, S.; Venkatraman, S.; Adams, E.L.; et al. Transcriptomic Profiling of the Developing Cardiac Conduction System at Single-Cell Resolution. Circ. Res. 2019, 125, 379–397.

>> Ref 19: Hoogaars, W.M.H.; Engel, A.; Brons, J.F.; Verkerk, A.O.; de Lange, F.J.; Wong, L.Y.E.; Bakker, M.L.; Clout, D.E.; Wakker, V.; Barnett, P.; et al. Tbx3 controls the sinoatrial node gene program and imposes pacemaker function on the atria. Genes Dev. 2007, 21, 1098–1112.

>> Ref 26: Linscheid, N.; Logantha, S.J.R.J.; Poulsen, P.C.; Zhang, S.; Schrölkamp, M.; Egerod, K.L.; Thompson, J.J.; Kitmitto, A.; Galli, G.; Humphries, M.J.; et al. Quantitative proteomics and single-nucleus transcriptomics of the sinus node elucidates the foundation of cardiac pacemaking. Nat. Commun. 2019, 10, 2889

>> Furthermore, one important aspect of our review is the synthesis of the relevant regulatory genes as derived from the body of transcriptomics studies. We discuss this on Lines 85-92 and 265-267.

It is important for the reader to understand how extrinsic signals can influence the synchronization of the pacemaker cells. Therefore, the authors should better explain the relationships between environment and this system.

>> We discuss the following environmental influences:

>> 1) Hypoxia, infection: section 3, Lines 105 - 146;

>> 2) The connection to pacemaker cell synchronization: Section 6, Lines 222 - 256.

>> 3) The autonomic nervous system (sympathetic and vagal nerves to be precise): Lines 180 - 183, 191 - 194, 215 - 220.

The authors did not explain how the significance of heart rate variability (HRV) and how it would affect the development of the fetus.

>> The focus of the present review is on the cardiac pacemaker cell development. The activity of this system does give rise to intrinsic HRV (iHRV) the significance of which we discuss on Lines 25-31 in the broader context of fetal HRV as the mainstay of noninvasive fetal monitoring. We return to the question of HRV significance in Section 6 where we discuss the impact of fetal hypoxia and infection on cardiac pacemaker cell development and synchronization in the context of HRV monitoring. Consequently, the questions of HRV significance and the relationship to fetal development are tackled while maintaining a focus on cardiac pacemaker cell development and iHRV.

The author should further develop the part on hypoxia and in particular the pathway that is activated in response to hypoxia.

>> In general, Section 3 of the review deals with the influences of infection and hypoxia taking approx. 50/50 space for discussion of each adversity. While we attempt to cite the existing relevant literature, we also note that there are presently more questions than there are answers in this field of investigation. We identify such questions hoping they will represent the next logical steps of fruitful research in the field. We deliberately abstain from reviewing the physiology of hypoxia or infection per se citing instead the relevant work as needed.

>> Specifically, we identify several insights into the mechanisms by which hypoxia may impact cardiac pacemaker cell synchronization on Lines 130- 140 (Ref. 27 - 29).

>> First, we note the remarkable resilience of these cells to prolonged hypoxia which appears to be due to a lower global metabolic demand compared to myocardial cells (Lines 133 - 137).

Ref. 27: Kohlhardt, M.; Mnich, Z.; Maier, G. Alterations of the excitation process of the sinoatrial pacemaker cell in the presence of anoxia and metabolic inhibitors. J. Mol. Cell. Cardiol. 1977, 9, 477–488.

Ref. 28: Senges, J.; Mizutani, T.; Pelzer, D.; Brachmann, J.; Sonnhof, U.; Kübler, W. Effect of hypoxia on the sinoatrial node, atrium, and atrioventricular node in the rabbit heart. Circ. Res. 1979, 44, 856–863.

Ref 29: Nishi, K.; Yoshikawa, Y.; Sugahara, K.; Morioka, T. Changes in electrical activity and ultrastructure of sinoatrial nodal cells of the rabbit’s heart exposed to hypoxic solution. Circ. Res. 1980, 46, 201–213.

>> Second, we bring in the notion of the mitochondrial dynamics as a key modulator of the energy availability and the cardiac pacemaker cell synchronization (Line 130, Ref. 20).

Ref. 20: Gu, J.-M.; Grijalva, S.I.; Fernandez, N.; Kim, E.; Foster, D.B.; Cho, H.C. Induced cardiac pacemaker cells survive metabolic stress owing to their low metabolic demand. Exp. Mol. Med. 2019, 51, 105.

Third, we cite the additional relevant literature for the interested reader in Ref. 2, 37, 38:

Ref. 2: Giussani, D.A.; Davidge, S.T. Developmental programming of cardiovascular disease by prenatal hypoxia. J. Dev. Orig. Health Dis. 2013, 4, 328–337

Ref. 37: Giussani, D.A.; Camm, E.J.; Niu, Y.; Richter, H.G.; Blanco, C.E.; Gottschalk, R.; Blake, E.Z.; Horder, K.A.; Thakor, A.S.; Hansell, J.A.; et al. Developmental programming of cardiovascular dysfunction by prenatal hypoxia and oxidative stress. PLoS One 2012, 7, e31017.

Ref. 38: Rouwet, E.V.; Tintu, A.N.; Schellings, M.W.M.; van Bilsen, M.; Lutgens, E.; Hofstra, L.; Slaaf, D.W.; Ramsay, G.; Le Noble, F.A.C. Hypoxia induces aortic hypertrophic growth, left ventricular dysfunction, and sympathetic hyperinnervation of peripheral arteries in the chick embryo. Circulation 2002, 105, 2791–2796.

Figure 1: The illustration is very complex and difficult to understand. The authors should simplify or better explain this Figure.

>> We appreciate the reviewer’s concern, especially since this figure is meant to serve as the visual abstract of the review. We have completely rewritten the legend to Figure 1 expanding the explanation of each component depicted therein and simplifying for better readability. The edits are indicated by the red font on Lines 398-436.

Reviewer 2 Report

General: most of the text focuses on rather fine variables, such s CPS and (i)HRV. However, simple HR (fetal bradycardia/tachycardia) should not be neglected, as it also indicates ongoing injury and has profound effects on fetal well-being (as reviewed recently in Acta Physiol 2015, 213, 303–320).

Specific comments

Line 1 end – brown adipose tissue – why not consider also glycogen, in which the primary myocardium is relatively rich?

Line 130-140: again, glycolytic (anaerobic) metabolism characteristic of primary myocardium (discussed extensively in the works of Eric Raddatz group) might be mentioned.

Line 165: make distinction between truly immature myocytes (tubular heart), chamber formation period (differential pacemaking properties, also vastly different levels of HCN4 expression), and postnatal stages. See e.g. Am J Physiol Regulatory Integrative Comp

Physiol 283: R379–R388, 2002. HCN4 is expressed in the embryonic chamber myocardium at low levels, making it a suitable marker of SAN already from pre-septation stages.

Line 269: how does the Langendorff model come into play? In common fetal models (mouse, rat), it is not really feasible as these hearts are too small – and in the adult animals are not exactly the same as feta. If the authors mean studying the impact of prenatal environment/insults on adult heart, it should be explicitly stated.

Author Response

General: most of the text focuses on rather fine variables, such s CPS and (i)HRV. However, simple HR (fetal bradycardia/tachycardia) should not be neglected, as it also indicates ongoing injury and has profound effects on fetal well-being (as reviewed recently in Acta Physiol 2015, 213, 303–320).

>> We thank the reviewer for this comment. We added the discussion of the effect of fetal adversity on HR such as fetal bradycardia/tachycardia citing [2] on Line 140 where we discuss the role of intrauterine adversity in arrhythmia development. We added this as Ref. 50 to the review.

Specific comments

Line 1 end – brown adipose tissue – why not consider also glycogen, in which the primary myocardium is relatively rich?

>> The literature reports the specific role of the brown adipose tissue for the sinoatrial node, hence our focus on this tissue. We now added a contrasting comparison to myocardium on Line 45-46.

Line 130-140: again, glycolytic (anaerobic) metabolism characteristic of primary myocardium (discussed extensively in the works of Eric Raddatz group) might be mentioned.

>> We thank the reviewer for this comment! We now added the citations of the work by this group: we cite [3] as Reference #52.

>> Specifically, in Section 3 dealing with the impact of hypoxia, on Lines 139-141 we mention specifically that embryonic development as a sensitive stage for intrauterine adversity.

>> We return to this notion in Section 5 dealing with iHRV, on Lines 210-214. Finally, in Section 7 (Conclusions) on Lines 262-266, we stress the importance of future research into embryonic and fetal stages of intrauterine development to identify periods vulnerable to adversity.

>> On Lines 16-21, we note that the exact timing of the emergence of cardiac pacemaker cell synchronization is not known.

Line 165: make distinction between truly immature myocytes (tubular heart), chamber formation period (differential pacemaking properties, also vastly different levels of HCN4 expression), and postnatal stages. See e.g. Am J Physiol Regulatory Integrative Comp Physiol 283: R379–R388, 2002. HCN4 is expressed in the embryonic chamber myocardium at low levels, making it a suitable marker of SAN already from pre-septation stages.

>> We thank the reviewer for the insightful comments. We now added the reference to the recommended study as [3] on Lines 139-141 (Ref #52 in our review) and make the distinction between truly immature myocytes, chamber formation period and the mature fetal stages.

>> In Section 4, on Lines 166-171, we denote the difference in the pacemaker cell differentiation status in the embryonic versus fetal development. We thank the reviewer for the addition of the useful information on the HCN4 expression and added this important insight on Lines 172-174.

Line 269: how does the Langendorff model come into play? In common fetal models (mouse, rat), it is not really feasible as these hearts are too small – and in the adult animals are not exactly the same as feta. If the authors mean studying the impact of prenatal environment/insults on adult heart, it should be explicitly stated.

>> We appreciate the reviewer’s comment that the small animal models of fetal development are not well suited for ex vivo heart studies using the Langendorff approach. This is why we refer in this review to the unique in vivo - ex vivo animal model using chronically instrumented fetal sheep followed by the Langendorff preparation (Lines 180-182, 197-203, Reference #36). We apologize for being not clear about the reference: it was in fact missing in this place. We now added the missing reference on Line 272.

>> Ref. 36: Frasch, M.G.; Herry, C.L.; Niu, Y.; Giussani, D.A. First evidence that intrinsic fetal heart rate variability exists and is affected by hypoxic pregnancy. J. Physiol. 2019.

Added references

  1. Krogh-Madsen, T.; Kold Taylor, L.; Skriver, A.D.; Schaffer, P.; Guevara, M.R. Regularity of beating of small clusters of embryonic chick ventricular heart-cells: experiment vs. stochastic single-channel population model. Chaos 2017, 27, 093929.
  2. Sedmera, D.; Kockova, R.; Vostarek, F.; Raddatz, E. Arrhythmias in the developing heart. Acta Physiol. 2015, 213, 303–320.

    3.         Sedmera, D.; Kucera, P.; Raddatz, E. Developmental changes in cardiac recovery from anoxia-reoxygenation. Am. J. Physiol. Regul. Integr. Comp. Physiol. 2002, 283, R379–88.